# Impact of circadian rhythm preference on performance in China's standardized residency training: The mediating role of loneliness

Yi Yang[1]☯, Yanxia Zhu[2]☯, Jianqiang Ni[1]☯, Jiang Yan[2], Jun Yin[3], Le Yang[4]*, Hao Li[5]*

1 Department of Neurology, The First Affiliated Hospital of Soochow University, Suzhou, Jiangsu, China, 2 Department of Education and Training Branch, The First Affiliated Hospital of Soochow University, Suzhou, Jiangsu, China, 3 Department of General Surgery, The First Affiliated Hospital of Soochow University, Suzhou, Jiangsu, China, 4 Center for Digital Intelligence in Medicine, Fuzhou University Affiliated Provincial Hospital, Fuzhou, Fujian, China, 5 Department of Neurology, The Fourth Affiliated Hospital of Soochow University, Suzhou, Jiangsu, China

☯ These authors contributed equally to this work.
* ylsqyh@163.com (LY); 18862156182@163.com (HL)

## Abstract

### Objective

This study aimed to clarify how circadian rhythm preference influences academic performance within the evolving framework of standardized residency training in China, where research on residents' psychological adaptation and social support remains limited. The main objective was to explore the relationship between individual chronotypes and standardized training performance, as well as the potential mediating effect of loneliness levels.

### Methods

Basic information of residents who completed standardized training at the First Affiliated Hospital of Soochow University was retrospectively collected, including baseline information such as gender, place of study, and program type. Participants' chronotypes and loneliness levels were evaluated using the Morningness-Eveningness Questionnaire-Self-Assessment and the University of California, Los Angeles Loneliness Scale. We then explored the relationships among chronotype, loneliness level, and academic performance using generalized linear modeling and mediation analysis.

### Results

A total of 198 participants who had completed their residency training were included, among whom 93 (47.0%) achieved relatively higher final scores. After adjusting for confounding factors, the generalized linear models revealed significant associations between MEQ-SA scale (OR: 0.811, 95% CI: 0.770–0.855, p < 0.001), UCLA

**Data availability statement:** Data cannot be shared publicly because they contain personally identifiable information of medical students. Data are available from the corresponding author or the Ethics Committee of the First Affiliated Hospital of Soochow University for researchers who meet the criteria for access to data: (Ximeng Zhang, 405415168@qq.com).

**Funding:** This work was supported by grants from the National Natural Science Foundation of China (82201446) and the Natural Science Foundation of Jiangsu Province of China (BK20220249).

**Competing interests:** The authors have declared that no competing interests exist.

Loneliness scale (OR: 1.793, 95% CI: 1.076–1.323, p = 0.001), and the standardized training grades. Furthermore, mediation analysis indicated that loneliness level acted as a mediator in the association between chronotypes and training achievements (mediating role = 55.66%).

## Conclusion

Unlike adolescents, residents in standardized training with an evening chronotype achieved higher final grades. Notably, loneliness significantly mediated this relationship, challenging conventional views on social isolation and learning. These findings underscore the need for further research to clarify underlying mechanisms and potential interventions in medical training.

---

## Introduction

The circadian rhythm is a biological regulator of physiological functions governed by the molecular clock, which plays a crucial role in human health and well-being [1]. It generates physiological and behavioral cycles every approximately 24 hours, aligning internal processes with the external environment [1]. Circadian rhythm disruption can seriously affect sleep-wake cycles, hormone secretion, eating patterns, and other basic bodily functions [2,3]. Recent studies have highlighted that circadian rhythms influence not only physiological and psychological aspects but also cognitive processes such as working memory, executive function, and attention [2–4]. An important aspect of circadian rhythms is chronotype, which refers to an individual's preference for morning or evening activities [5].

Loneliness, as a complex and pervasive individual characteristic and emotional state, has been shown to affect various aspects of personal well-being, including mental health, social relationships, and overall life satisfaction [6,7]. Notably, Notably, previous research suggests that loneliness can negatively impact cognitive functioning, motivation, and academic performance, particularly in student populations, where social connectedness is crucial for educational success [7].

Meanwhile, a growing body of evidence confirms the close relationship between circadian rhythms and individual characteristics of loneliness. Current studies have focused on the combined effects of these two factors on pathogenesis, emphasizing the important roles of loneliness and chronotype in cognitive and emotional functioning [8–10]. However, the impact of circadian rhythm preferences and loneliness characteristics on academic performance remains unclear. This is particularly relevant in residency training programs, where residents contend with the dual pressures of academic and clinical responsibilities, potentially disrupting their natural sleep patterns [11]. Additionally, the rotation system frequently alters their social connections over short periods, which can exacerbate feelings of isolation and further influence their academic performance [12]. Therefore, there is a need for further investigation into how circadian rhythms and loneliness affect medical students' performance during residency training.

The purpose of this study was to examine the multifactorial interactions between circadian rhythm preferences and loneliness levels in relation to final grades in a residency training program. As China's standardized residency training system has been implemented for just over a decade, research on its impact remains limited, particularly regarding the psychological changes residents experience and the social support they receive during this process. By objectively assessing the combined effects of loneliness levels and chronotype preference on residency training program outcomes, this study will contribute to filling this research gap, informing the development of preventive interventions, and providing valuable guidance for educational practice.

## Methods

### Setting

Considering the contextual framework, our institution, the First Hospital Affiliated to Soochow University, operates across two campuses in Suzhou and hosts residents from all over the country who participate in standardized training programs. The residency program is divided into two categories. The first is a combined training program integrated with postgraduate studies, where students enroll directly into the residency program and must fulfill both postgraduate coursework requirements and residency training requirements upon graduation. The second category is a socialized residency training program, designed for residents who have not previously participated in standardized training. Both types of programs span three years, with clinical training being largely consistent between them.

Each program is tailored to the resident's specialty, with rotational training lasting approximately 1–3 months per specialty. Postgraduate students also have additional coursework and research training requirements. The primary language of instruction is Chinese, though some components are delivered in English. In the clinical setting, feedback is mainly provided verbally, supplemented by mandatory written monthly evaluations for each rotation and exit evaluations at each site. These evaluations are accessible to residents for their final assessments. Each clinical site has two co-supervisors who offer feedback to the residents participating in the program.

### Sample and procedures

Between March 2024 and June 2024, we distributed questionnaires to residents who completed standardized training from the 2019–2021 cohort at each campus. The questionnaires were collected between September 28, 2024, and October 28, 2024. They were disseminated through lecture announcements, email, and an applet. Our goal was to achieve a diverse sample, ensuring a balanced representation of participants across gender, study location, program type, and training specialty. All residents routinely wrote electronic medical records for both the inpatient beds under their management in the corresponding wards and for outpatient/emergency encounters. The number of inpatient beds managed by each resident remained relatively consistent (3–4 beds). The frequency of electronic classroom use was generally fixed, and residents were required to attend multiple academic lectures each month (commonly around five). The number of monthly night shifts was broadly consistent across participants, usually ranging from 4+2 to 5+2 shifts, with only minor variations between individuals. These comparable schedules indicate that participants were likely to experience similar levels of sleep disruption and evening exposure to blue light. Furthermore, all residents lived either in hospital staff dormitories or in apartments within approximately 1 km of the hospital, suggesting that their levels of exposure to outdoor nighttime lighting were also similar. All residents used smartphones; however, only two participants occasionally used the night (dark) mode feature, which was insufficient to conduct stratified analysis. We recorded the time taken to complete the questionnaire and excluded all responses from participants who took less than ninety seconds to complete it. Subjects with known emotional abnormalities, severe sleep disorders, or orofacial disorders, as well as those who did not wish to participate in the study, were also excluded.

The study was approved by the Ethics Committee of the First Affiliated Hospital of Soochow University before the questionnaire collection began. The experiments comply with the current laws of the country in which they were performed. At

the beginning of each electronic questionnaire, participants were presented with a detailed informed consent form outlining the study's purpose, procedures, potential risks and benefits, as well as their right to withdraw at any time without penalty. Participants provided explicit written informed consent by agreeing to this form. All collected data were anonymized and treated with strict confidentiality to ensure privacy.

### Measurement

In addition to gathering basic demographic information, the survey incorporated two key assessment tools. Chronotype was evaluated using the Morningness-Eveningness Questionnaire-Self-Assessment (MEQ-SA), which comprises 19 items aimed at determining an individual's inclination towards morning or evening activities [13]. Responses are scored on either a 4- or 5-point Likert scale, depending on the specific question, resulting in a total score ranging from 16 to 86. Higher scores reflect a stronger morning preference, whereas lower scores indicate a tendency towards eveningness. The University of California, Los Angeles Loneliness Scale (UCLA-LS), third edition (1988), was used to assess the overall level of loneliness among participants. This scale consists of 20 items, divided equally between positive and negative statements, with responses recorded on a 4-point scale, where 1 represents "never" and 4 represents "frequently." The total score can range from 20 to 80, with higher scores correlating with greater perceived loneliness. The Chinese version of the UCLA-LS has been previously validated and demonstrated consistent internal reliability with Chinese college students [14,15]. Each participant was also asked a single question: "What is the one goal you most want to achieve during the three-year residency training?" They were instructed to answer with a single sentence or phrase. Furthermore, data on normative training completion scores were retrieved and matched using an electronic system.

### Data analysis

Data normality was assessed using the Kolmogorov-Smirnov test. Since continuous variables showed non-normal distributions, non-parametric tests were employed for all analyses. For descriptive analysis, differences between continuous variables were evaluated using the Mann-Whitney U-test, while differences in categorical variables were assessed with the chi-square test.

Spearman correlation coefficients were computed to examine the relationships among the variables. The relationship between chronotype, loneliness levels, and final training completion scores was analyzed using a generalized linear model with binomial family and logit link function to analyze the binary outcome variable (high vs. low grades), adjusting for general demographic factors (gender and age).

To further investigate whether loneliness acts as a mediating factor in the relationship between chronotype and training outcomes, mediation analysis was conducted using the R "mediation" package with 1,000 bootstrap iterations to explore the indirect role of loneliness. Average causal mediation effect (ACME), average direct effect (ADE), and proportion mediated were calculated for pathways.

All statistical analyses were performed using Origin 2024, Vosviewer 1.6.11 and R version 4.3.2, with a two-sided significance level set at 0.05. To address multiple testing concerns, p-values from multiple comparisons in the GLM were adjusted using the False Discovery Rate (FDR) correction method.

## Results

### Characteristics of participants

A total of 198 participants who had completed their residency training were included in our study. They were divided into two groups based on a median cutoff score of 429, with 93 participants achieving relatively higher final scores. No significant differences were observed between the two groups regarding gender and age. However, there were significant differences in chronotype and loneliness levels (p < 0.001). Among all participants, 79 (39.9%) identified themselves as having

an intermediate chronotype. Notably, in the higher-performing group, 41 (44.1%) were categorized as evening-type, compared to only 2 (1.9%) in the lower-performing group (*p* < 0.05; Table 1).

In addition, correlation coefficients were calculated between grade level, gender, and scores on each question of the MEQ-SA scale and the UCLA Loneliness Scale. The results showed that there was a significant correlation between the standardized training closing score and the type of time and the level of loneliness (Fig 1).

### Effect of chronotype and loneliness level on grades

A bubble chart was utilized to visually represent the distribution of chronotype, loneliness levels, and grades (Fig 2). The effects of chronotype and loneliness levels on standardized training grades were further analyzed by using generalized linear models. Further multivariate analysis was performed to evaluate the effects of chronotype and loneliness levels on standardized training grades. All reported p-values were adjusted for multiple comparisons using the false discovery rate (FDR) correction method. For the overall sample, in multivariate analyses, we found that the MEQ-SA scale (OR: 0.811, 95% CI: 0.770–0.855, *p* < 0.001) and UCLA Loneliness scale (OR: 1.193, 95% CI: 1.076–1.323, *p* = 0.001) was significantly associated with the standardized training grades after adjusting for age and gender (Table 2). Chronotypes were categorized based on scale scores, with scores ranging from 42 to 58 classified as "intermediate type," while the remaining participants were classified as either "evening type" or "morning type." Our analysis revealed that the evening type (OR: 15.431, 95% CI: 3.431–69.395, p < 0.001) and morning type (OR: 0.057, 95% CI: 0.022–0.149, p < 0.001) were significantly associated with standardized training grades after adjusting for age and gender (Table 2).

**Table 1. Characteristics of participants\*.**

| Variable | Low grades | High grades | *P*‡ |
|---|---|---|---|
| | N = 105 | N = 93 | |
| Age in years, n (%) | | | 0.621 |
| 20-25, n (%) | 10 (9.52%) | 13 (14.0%) | |
| 25-30, n (%) | 75 (71.4%) | 63 (67.7%) | |
| 30-35, n (%) | 20 (19.0%) | 17 (18.3%) | |
| Gender | | | 0.696 |
| Female, n (%) | 57 (54.3%) | 54 (58.1%) | |
| Male, n (%) | 48 (45.7%) | 39 (41.9%) | |
| MEQ-SA scale | | | <0.001 |
| Eveningness | 2 (1.90%) | 41 (44.1%) | |
| Intermediate | 33 (31.4%) | 46 (49.5%) | |
| Morningness | 70 (66.7%) | 6 (6.45%) | |
| UCLA Loneliness scale in socres, median(IQR) | 6.00 (4.00; 8.00) | 8.00 (6.00; 9.00) | <0.001 |
| MEQ-SA scale in socres, median(IQR) | 60.0 (57.0; 64.0) | 42.0 (40.0; 47.0) | <0.001 |
| Achievements of resident standardisation training in socres, median(IQR) | 420 (413; 425) | 439 (435; 445) | <0.001 |

Abbreviations: MEQ-SA: Morningness-Eveningness Questionnaire-Self-Assessment; UCLA: University of California, Los Angeles (Loneliness Scale); IQR: Interquartile Range.

\* Data are presented as absolute numbers (percentages) for categorical variables and median (first quartile–third quartile) for continuous variables with skewed data.

‡ Chi-squared test for categorical variables and Mann–Whitney *U* test for skewed continuous variables.

*P* < 0.05 was considered statistically significant.

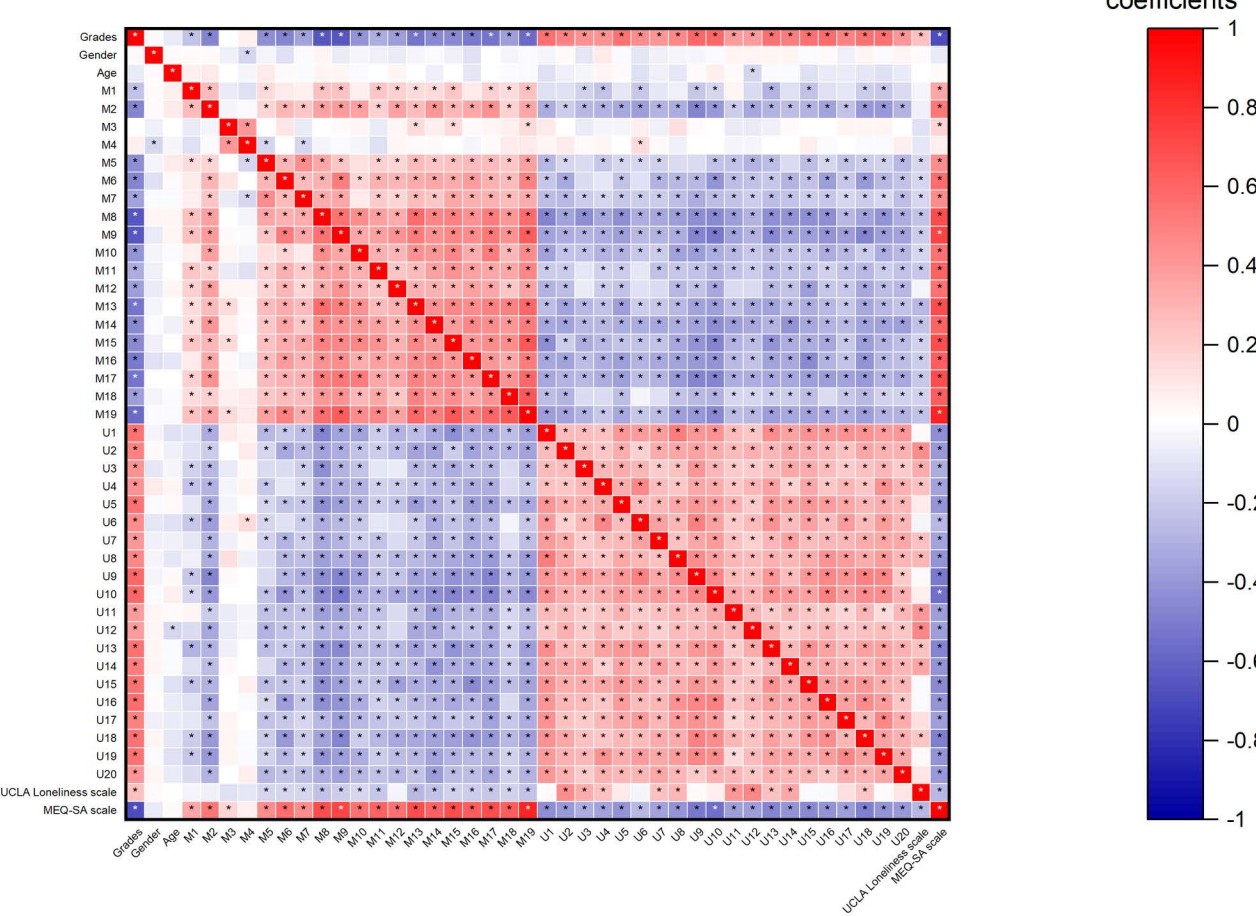

* p<=0.05

**Fig 1. The heat map for pearson correlation coefficients between the variables.**

## Mediating role of loneliness level

Our findings indicated that lower scores on the MEQ-SA scale were associated with a higher likelihood of achieving better exit grades in residency training, while higher scores on the UCLA Loneliness Scale also correlated with improved grades (Fig 3A). To further explore this relationship, a mediation analysis was conducted to determine whether loneliness level mediates the link between chronotype and success in standardized residency training (Fig 3B). As anticipated, loneliness level was identified as a significant mediator in this association. The indirect effect of loneliness level on the association between chronotypes and training achievements implied a 0.0186-point increase in the likelihood of Better grades (95% CI, 0.0077–0.0279), accounting for approximately 55.66% of the association between chronotypes and achievements of resident standardisation training (Fig 3B).

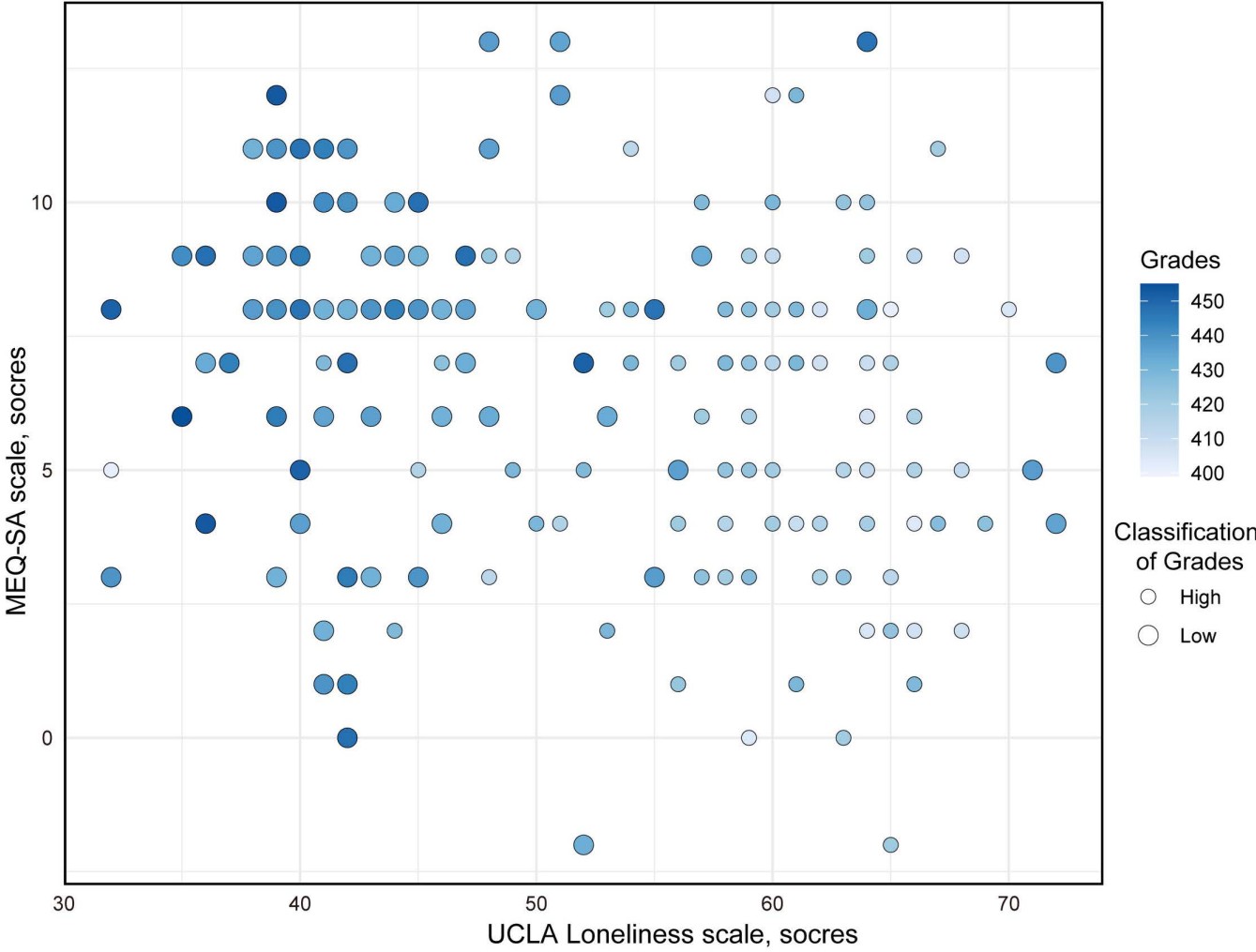

**Fig 2. The bubble chart for the distribution of chronotype, loneliness levels, and grades.**

## Clustering of text

The study included responses from 198 participants to the question, "What is the one goal you most want to achieve during the three-year residency training?" Keywords were derived from participants' answers and displayed through a text clustering model (Fig 4). In this model, terms such as "standard training," "graduation," "future," "exam," "work," "stay," and "article" appeared frequently in the responses, with "graduation" being the most common keyword, apart from "standard training."

## Discussion

In this study, we performed a comprehensive analysis to explore the interactions between chronotype, loneliness levels, and performance in standardized residency training. The principal findings of this study can be summarized as follows: (1) Evening chronotypes and higher loneliness levels were linked to better grades; (2) loneliness level played a mediating role in the success in standardized residency training, indicating a potential connection between psychosocial well-being and academic performance.

**Table 2. Association between chronotype, loneliness level, and grades.**

| Variable | Univariate OR (95% CI) | P | Multivariate OR (95% CI) | P | Adjusted P* |
|---|---|---|---|---|---|
| MEQ-SA scale, per 1-score increase | 0.816 (0.776, 0.858) | <0.001 | 0.811 (0.77, 0.855) | <0.001 | <0.001 |
| Categories | | | | | |
| Intermediate | Ref | <0.001 | Ref | <0.001 | <0.001 |
| Eveningness | 14.707 (3.321, 65.127) | <0.001 | 15.431 (3.431, 69.395) | <0.001 | <0.001 |
| Morningness | 0.061 (0.024, 0.158) | <0.001 | 0.057 (0.022, 0.149) | <0.001 | <0.001 |
| UCLA Loneliness scale, per 1-score increase | 1.195 (1.078, 1.324) | <0.001 | 1.193 (1.076, 1.323) | <0.001 | 0.001 |

Abbreviations: *OR*: odds ratio; *CI*: confidence interval; Ref: reference.

Adjusted for age and gender in multivariate analysis.

*: *P*-values adjusted by using the FDR multiple correction method.

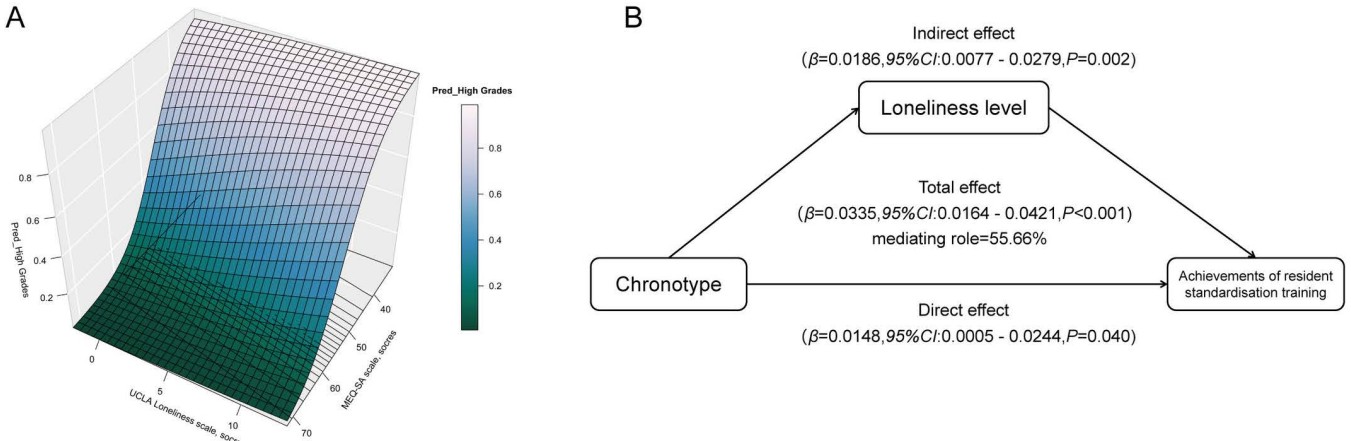

**Fig 3. Mediating role of loneliness level. (A)**: 3D Surface Plot for the relationship between chronotype, loneliness levels, and academic performance. **(B)** Mediated associations between chronotype, loneliness levels, and achievements of resident standardisation training. Abbreviations: *OR*: odds ratio; *CI*: confidence interval. Adjusted for age and gender in mediation analysis.

Certain aspects of residency training offer a unique framework for exploring the relationship between chronotype and performance [16]. Compared to the general student population, standardized training residents typically have less flexibility in setting sleep and wake-up times, making them particularly vulnerable to the demands of shift work and nighttime study. This lack of flexibility may significantly impact their wake-up times and overall psychosocial adjustment. Consequently, using samples of standardized training physicians to examine associations between early and late risers, academic performance, and loneliness levels offers valuable insights.

Our findings indicate that most participants were categorized as intermediate chronotypes, followed by early morning and, to a lesser extent, evening chronotypes. This distribution differs slightly from previous studies on undergraduate medical students, which may be attributed to the standardized training and the gradual increase in early risers with age [5,17,18]. While earlier studies often found that early morning types performed better academically, our results challenge

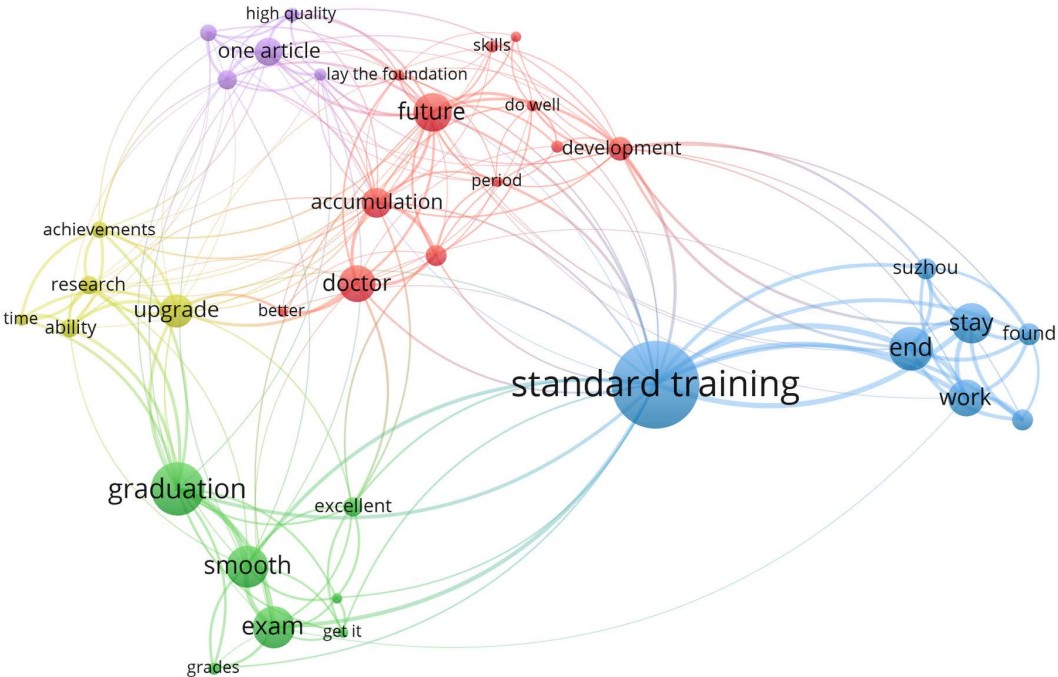

**Fig 4. Text clustering of participants' responses to their primary goal during residency training.** Each dot represents a keyword, with dot size indicating its frequency, and different colors denoting clusters.

this notion [19]. Notably, most previous studies focused on adolescents under 22 years of age, relying on younger samples, where outcomes were influenced by various factors [5,17–19]. Research suggests that evening chronotypes are typically associated with poorer task management skills and self-control [20–22]. However, residents in this study were generally high achievers with strong self-management skills, and all were older than 22 years (post-undergraduate). Previous research has shown that the effect of evening chronotype on grades extends beyond self-management issues, with social jet lag playing a critical role [23]. In the context of residency training, where most theoretical study time occurs in the evening, adjustments related to social jet lag become even more pronounced. Additionally, the frequent shift work may desynchronize circadian rhythms, further influencing self-rated chronotypes [24]. It is also likely that these self-assessed chronotypes are closely tied to adjustments in sleep debt during rest days [25]. Whether these observed chronotype shifts are due to stress-induced sleep alterations, changes in self-perception, or genuine chronotype changes, further research is required to elucidate the underlying mechanisms.

A growing body of evidence suggests that external factors such as social engagement and the degree of social isolation are significant influences on chronotype [26]. Social isolation, often considered a specific aspect of loneliness, is characterized by a lack of social networks [27]. In the unique context of standardized residency training, residents face a constantly changing rotation schedule, with each individual spending 1–3 months in different subspecialties. This frequent rotation means that each month brings new colleagues and faculty, and many residents do not live in shared housing. As a result, their social networks are disrupted regularly, making it challenging to establish consistent social connections, which may significantly contribute to social isolation.

Evidence suggests that loneliness levels may contribute to sleep fragmentation and poor sleep quality [28]. Previous research has shown that declining levels of loneliness, particularly social loneliness, in adolescents are strongly linked to the development of evening chronotype [29]. Additionally, it has been reported that social rejection triggers a stress

response, elevating cortisol levels, which in turn can cause sleep problems, such as delayed sleep onset and sleep disturbances [30]. Prior findings also indicate that stress exposure during puberty can dampen hypothalamic-pituitary-adrenal axis activity, potentially altering circadian rhythms through changes in sleep-related hormones [31]. Given the high-stress environment of residency training, this may explain the mediating role of loneliness levels between chronotype and academic performance, highlighting the need for psychological and social support for residents.

In interpreting our findings, it is important to note that we attempted to minimize potential confounding by ensuring that participants had similar schedules in terms of night shifts, lecture attendance, living arrangements, and access to electronic devices. Despite this control, higher levels of loneliness—particularly social loneliness—may involve reduced engagement in social activities, thereby freeing up more time for studying. This possible pathway could partly explain the positive association observed between loneliness and academic performance in our study. Prior research also suggests that stability in levels of social isolation over time may buffer against the negative effects of isolation [32,33]. This is because the longer adolescents remain isolated, the more they may adapt to being alone, potentially even developing positive attitudes toward this state [32,33]. The positive correlation between loneliness levels and academic performance observed in our study aligns with these findings. Nevertheless, because our study did not measure actual study time, this interpretation should be viewed with caution and verified in future research.

There are several limitations to this study. First, the generalizability of our findings may be restricted because all participants were residents from the same hospital. Therefore, applying our findings to populations with different characteristics should be done with caution. Second, we did not assess sleep duration or study time, which may partly explain the association between loneliness and academic performance. Third, despite conducting mediation analyses, our capacity to analyze various aspects of chronotype and loneliness levels was limited. Future research should explore different dimensions of chronotype and loneliness by tracking sleep through objective measures and recording loneliness trajectories, sleep quality, and social jet lag more accurately. Lastly, the design of this study does not allow for direct inference of a causal relationship between loneliness levels and chronotype. Further investigation is required.

## Conclusion

In conclusion, our data reveal a significant yet complex relationship between evening chronotype, higher levels of loneliness, and better academic performance among residents in standardized training programs. Notably, our findings suggest that loneliness may play a mediating role in this association, raising important questions about the underlying mechanisms. While these results challenge conventional perspectives on the impact of social isolation and circadian rhythm preferences on academic success, they also underscore the need for further research to clarify the broader implications for resident well-being and training outcomes. Understanding these dynamics could inform policies aimed at optimizing both performance and mental health in medical training.

## Author contributions

**Conceptualization:** Yi Yang, Le Yang, Hao Li.

**Data curation:** Jianqiang Ni, Jun Yin, Hao Li.

**Formal analysis:** Yi Yang, Le Yang, Hao Li.

**Funding acquisition:** Yi Yang.

**Investigation:** Yanxia Zhu, Jianqiang Ni.

**Methodology:** Yi Yang.

**Project administration:** Hao Li.

**Resources:** Jiang Yan.

**Software:** Yi Yang.

**Supervision:** Jiang Yan, Hao Li.

**Validation:** Yanxia Zhu.

**Writing – original draft:** Yi Yang, Yanxia Zhu, Le Yang.

**Writing – review & editing:** Hao Li.

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
