## [Decision Letter · Decision Letter 0]

15 Jul 2025

PONE-D-25-16590Impact of Circadian Rhythm Preference on Performance in China's Standardized Residency Training: The Mediating Role of LonelinessPLOS ONE

Dear Dr. Yang,

Thank you for submitting your manuscript to PLOS ONE. After careful consideration, we feel that it has merit but does not fully meet PLOS ONE’s publication criteria as it currently stands. Therefore, we invite you to submit a revised version of the manuscript that addresses the points raised during the review process.

**The reviewer raised a number of questions and points to address. I would expect that you respond to each of these. It is particularly important to provide sufficient detail regarding the statistical methods used and potential issues with confounding variables multiple testing correction. The interpretation of the data should mention the relationship between evening chronotypes and light exposure from artificial sources as well as sleep deprivation and a possible impact of loneliness on study time.**  In addition, it may be worth considering whether extensive studying in front of a computer screen at night could induce an evening chronotype as well as increase the risk of loneliness.

We look forward to receiving your revised manuscript.

Kind regards,

Herman Wijnen, Ph.D.

Academic Editor

PLOS ONE

Journal Requirements:

This work was supported by grants from the National Natural Science Foundation of China (82201446) and the Natural Science Foundation of Jiangsu Province of China (BK20220249).

6. We note that you have indicated that there are restrictions to data sharing for this study. For studies involving human research participant data or other sensitive data, we encourage authors to share de-identified or anonymized data. However, when data cannot be publicly shared for ethical reasons, we allow authors to make their data sets available upon request. For information on unacceptable data access restrictions, please see http://journals.plos.org/plosone/s/data-availability#loc-unacceptable-data-access-restrictions.

Reviewers' comments:

Reviewer's Responses to Questions

**Comments to the Author**

1. Is the manuscript technically sound, and do the data support the conclusions?

Reviewer #1: Partly

2. Has the statistical analysis been performed appropriately and rigorously? 

Reviewer #1: No

3. Have the authors made all data underlying the findings in their manuscript fully available?

Reviewer #1: Yes

4. Is the manuscript presented in an intelligible fashion and written in standard English?

Reviewer #1: Yes

5. Review Comments to the Author

Reviewer #1: This is an interesting and potentially valuable investigation of the relationship of circadian rhythm preference (chronotype), loneliness, and academic performance in Chinese medical residents undergoing standardized training. The findings have the potential to inform interventions and support programs in order to improve the well-being and academic success of residents undergoing standardized training. I believe this work can be recommended, however, there are several crucial issue that have to be addressed first:

1. Considering the increasing prevalence of smartphone use and its potential impact on circadian rhythms via blue light exposure, did you investigate whether differences in smartphone usage patterns or evening light exposure among your participants might have influenced the observed relationship between chronotype, loneliness, and academic performance?

2. The study found that evening chronotype residents achieved higher final grades, a contrast to findings in adolescents. To better understand these differing outcomes, did the authors consider the potential influence of differences in study time or learning strategies employed by residents with evening chronotypes? Furthermore, given that adolescents with evening chronotypes often experience chronic sleep restriction due to early school start times, did the authors explore whether a similar sleep debt and associated daytime impairments might contribute to the lower academic performance typically observed in this population? As sleep duration was not assessed in this study, lack of such crucial information has to be discussed as limitation.

3. The study proposes that loneliness mediates the relationship between chronotype and academic success. However, is it possible that loneliness, in and of itself, primarily serves as a factor enabling increased study time, thereby indirectly contributing to higher grades? Could this alternative explanation, where loneliness facilitates longer study hours, influence the interpretation of the observed mediating role?

4. Methods sections and specifically statistics requires improvements. Considering the limited detail provided regarding normality testing, model selection for the generalized linear model, and potential confounding variables in the mediation analysis, could the authors provide a more explicit description of these methodological aspects to enhance the rigor and transparency of their findings? Additionally, given the number of statistical tests performed, did the authors consider adjustments for multiple comparisons, and if not, how might this impact the interpretation of the reported results?

5. Considering that age is known to influence both chronotype and loneliness, I was wondering if the authors could comment on whether they assessed age as a potential confounding or moderating variable in their study? Any insights into the role of age in these relationships would be very helpful.

6. PLOS authors have the option to publish the peer review history of their article (what does this mean? ). If published, this will include your full peer review and any attached files.

**Do you want your identity to be public for this peer review?** For information about this choice, including consent withdrawal, please see our Privacy Policy .

Reviewer #1: No

---

## [Author Response · Author response to Decision Letter 1]

4 Aug 2025

Dear editor,

Thank you and expert reviewers for the very constructive comments on our manuscript. In this revision, we have provided more information and addressed the concerns from reviewers point by point. The comments really helped improve the quality of our manuscript.

Journal Requirements:

1.Please ensure that your manuscript meets PLOS ONE's style requirements, including those for file naming.

Response: We appreciate the editor’s reminder. The manuscript and file names have been revised to meet PLOS ONE’s style requirements as requested.

2.Please provide additional details regarding participant consent. In the ethics statement in the Methods and online submission information, please ensure that you have specified what type you obtained (for instance, written or verbal, and if verbal, how it was documented and witnessed). If your study included minors, state whether you obtained consent from parents or guardians. If the need for consent was waived by the ethics committee, please include this information.

Response: We appreciate the editor’s comment. We have revised the Methods section and the Ethics statement in the manuscript to include additional details regarding participant consent, specifying the type of consent obtained and providing the relevant information as requested. The revised section is as follows

Sample and procedures

The study was approved by the Ethics Committee of the First Affiliated Hospital of Soochow University before the questionnaire collection began. The experiments comply with the current laws of the country in which they were performed. At the beginning of each electronic questionnaire, participants were presented with a detailed informed consent form outlining the study's purpose, procedures, potential risks and benefits, as well as their right to withdraw at any time without penalty. Participants provided explicit written informed consent by agreeing to this form. All collected data were anonymized and treated with strict confidentiality to ensure privacy.

Ethics approval and consent to participate

This study was reviewed and approved by the Institutional Review Board of the First Affiliated Hospital of Soochow University. The experiments comply with the current laws of the country in which they were performed. The procedures used in this study adhere to the tenets of the Declaration of Helsinki. Informed consent was obtained from all participants prior to their inclusion in the study. Specifically, participants provided an electronic version of written informed consent, acknowledging that they had been fully informed about the study’s purpose, procedures, potential risks, and benefits. They were also informed of their right to withdraw from the study at any time without any penalty or loss of benefits. All data collected were handled confidentially and anonymized to ensure privacy.

3.PLOS requires an ORCID iD for the corresponding author in Editorial Manager on papers submitted after December 6th, 2016. Please ensure that you have an ORCID iD and that it is validated in Editorial Manager. To do this, go to ‘Update my Information’ (in the upper left-hand corner of the main menu), and click on the Fetch/Validate link next to the ORCID field. This will take you to the ORCID site and allow you to create a new iD or authenticate a pre-existing iD in Editorial Manager.

Response: Thank you for the reminder. The ORCID iD for the corresponding author has been created and successfully validated in Editorial Manager as requested.

4.Thank you for stating the following financial disclosure: This work was supported by grants from the National Natural Science Foundation of China (82201446) and the Natural Science Foundation of Jiangsu Province of China (BK20220249). Please state what role the funders took in the study. If the funders had no role, please state: "The funders had no role in study design, data collection and analysis, decision to publish, or preparation of the manuscript." If this statement is not correct you must amend it as needed. Please include this amended Role of Funder statement in your cover letter; we will change the online submission form on your behalf.

Response: Thank you for your comment. We have added the statement “The funders had no role in study design, data collection and analysis, decision to publish, or preparation of the manuscript.” to the Financial Disclosure section as requested.

5.Your ethics statement should only appear in the Methods section of your manuscript. If your ethics statement is written in any section besides the Methods, please delete it from any other section.

Response: We confirm that the ethics statement now appears only in the Methods section of the manuscript, as requested.

6.We note that you have indicated that there are restrictions to data sharing for this study. For studies involving human research participant data or other sensitive data, we encourage authors to share de-identified or anonymized data. However, when data cannot be publicly shared for ethical reasons, we allow authors to make their data sets available upon request. For information on unacceptable data access restrictions, please see http://journals.plos.org/plosone/s/data-availability#loc-unacceptable-data-

access-restrictions. Before we proceed with your manuscript, please address the following prompts: a) If there are ethical or legal restrictions on sharing a de-identified data set, please explain them in detail (e.g., data contain potentially identifying or sensitive patient information, data are owned by a third-party organization, etc.) and who has imposed them (e.g., a Research Ethics Committee or Institutional Review Board, etc.). Please also provide contact information for a data access committee, ethics committee, or other institutional body to which data requests may be sent. b) If there are no restrictions, please upload the minimal anonymized data set necessary to replicate your study findings to a stable, public repository and provide us with the relevant URLs, DOIs, or accession numbers. Please see http://www.bmj.com/content/340/bmj.c181.long for guidelines on how to de-identify and prepare clinical data for publication. For a list of recommended repositories, please see https://journals.plos.org/plos

one/s/recommended-repositories. You also have the option of uploading the data as Supporting Information files, but we would recommend depositing data directly to a data repository if possible. Please update your Data Availability statement in the submission form accordingly.

Response: Thank you for your comment. Due to ethical restrictions imposed by the Institutional Review Board of the First Affiliated Hospital of Soochow University, the data for this study cannot be publicly shared because they contain potentially identifying or sensitive participant information. As requested, we have revised the Data Availability statement in the manuscript and submission form to reflect this and have provided contact details for data access requests.

7.If the reviewer comments include a recommendation to cite specific previously published works, please review and evaluate these publications to determine whether they are relevant and should be cited. There is no requirement to cite these works unless the editor has indicated otherwise. 

Response: Thank you for the reminder. We confirm that no such citations have been included in the manuscript.

Reviewer #1:

Concern 1: Considering the increasing prevalence of smartphone use and its potential impact on circadian rhythms via blue light exposure, did you investigate whether differences in smartphone usage patterns or evening light exposure among your participants might have influenced the observed relationship between chronotype, loneliness, and academic performance?

Response: We sincerely appreciate your insightful comment. In our study design, we took steps to minimize the potential influence of environmental and behavioral confounders on the observed relationships between chronotype, loneliness, and academic performance. Specifically, we collected information on participants’ smartphone usage patterns, place of residence, night shift frequency, and lecture attendance frequency. All residents had similar night shift schedules and comparable lecture requirements, and they lived either in staff dormitories within the hospital or in apartments located within 1 km of the hospital. Such arrangements helped reduce variability in exposure to artificial outdoor light at night. Regarding smartphone use, only a very small proportion of participants reported enabling the night mode, making it difficult to conduct a meaningful statistical analysis for this factor. Consequently, no statistical analysis was performed for night mode usage. We have revised the Methods section to include these additional details. The revised section is as follows

Sample and procedures

Between March 2024 and June 2024, we distributed questionnaires to residents who completed standardized training from the 2019–2021 cohort at each campus. The questionnaires were collected between September 28, 2024, and October 28, 2024. They were disseminated through lecture announcements, email, and an applet. Our goal was to achieve a diverse sample, ensuring a balanced representation of participants across gender, study location, program type, and training specialty. All residents routinely wrote electronic medical records for both the inpatient beds under their management in the corresponding wards and for outpatient/emergency encounters. The number of inpatient beds managed by each resident remained relatively consistent�3-4 beds�. The frequency of electronic classroom use was generally fixed, and residents were required to attend multiple academic lectures each month (commonly around five). The number of monthly night shifts was broadly consistent across participants, usually ranging from 4+2 to 5+2 shifts, with only minor variations between individuals. These comparable schedules indicate that participants were likely to experience similar levels of sleep disruption and evening exposure to blue light. Furthermore, all residents lived either in hospital staff dormitories or in apartments within approximately 1 km of the hospital, suggesting that their levels of exposure to outdoor nighttime lighting were also similar. All residents used smartphones; however, only two participants occasionally used the night (dark) mode feature, which was insufficient to conduct stratified analysis. We recorded the time taken to complete the questionnaire and excluded all responses from participants who took less than ninety seconds to complete it. Subjects with known emotional abnormalities, severe sleep disorders, or orofacial disorders, as well as those who did not wish to participate in the study, were also excluded.

Concern 2: The study found that evening chronotype residents achieved higher final grades, a contrast to findings in adolescents. To better understand these differing outcomes, did the authors consider the potential influence of differences in study time or learning strategies employed by residents with evening chronotypes? Furthermore, given that adolescents with evening chronotypes often experience chronic sleep restriction due to early school start times, did the authors explore whether a similar sleep debt and associated daytime impairments might contribute to the lower academic performance typically observed in this population? As sleep duration was not assessed in this study, lack of such crucial information has to be discussed as limitation.

Response: We sincerely thank the reviewer for this thoughtful and constructive comment. We fully agree that differences in study time, learning strategies, and sleep duration could be important factors influencing the relationship between chronotype and academic performance, and that these aspects may also help explain discrepancies between our findings and those in adolescents. As noted, our study did not assess sleep duration, which is indeed a key limitation. We have now revised the Limitations section to explicitly acknowledge the lack of assessment of sleep duration and study time, and to suggest that future research should address these variables to better elucidate their potential impact. The revised section is as follows

There are several limitations to this study. First, the generalizability of our findings may be restricted because all participants were residents from the same hospital. Therefore, applying our findings to populations with different characteristics should be done with caution. Second, we did not assess sleep duration or study time, which may partly explain the association between loneliness and academic performance. Third, despite conducting mediation analyses, our capacity to analyze various aspects of chronotype and loneliness levels was limited. Future research should explore different dimensions of chronotype and loneliness by tracking sleep through objective measures and recording loneliness trajectories, sleep quality, and social jet lag more accurately. Lastly, the design of this study does not allow for direct inference of a causal relationship between loneliness levels and chronotype. Further investigation is required.

Concern 3: The study proposes that loneliness mediates the relationship between chronotype and academic success. However, is it possible that loneliness, in and of itself, primarily serves as a factor enabling increased study time, thereby indirectly contributing to higher grades? Could this alternative explanation, where loneliness facilitates longer study hours, influence the interpretation of the observed mediating role?

Response: We thank the reviewer for this insightful comment and agree that loneliness could potentially serve as a factor enabling increased study time, thereby indirectly contributing to higher grades. We have revised the Discussion section to reflect this alternative explanation and to clarify that, although we attempted to minimize potential confounding by ensuring similar schedules in terms of night shifts, lecture attendance, living arrangements, and access to electronic devices, higher levels of loneliness—particularly social loneliness—may still result in reduced engagement in social activities, thus allowing more time for studying. We have also noted that, since our study did not measure actual study time, this interpretation should be considered with caution and explored in future research. The revised paragraph now reads as follows:

In interpreting our findings, it is important to note that we attempted to minimize potential confounding by ensuring that participants had similar schedules in terms of night shifts, lecture attendance, living arrangements, and access to electronic devices. Despite this control, higher levels of loneliness—particularly social loneliness—may involve reduced engagement in social activities, thereby freeing up more time for studying. This possible pathway could partly explain the positive association observed between loneliness and academic performance in our study. Prior research also suggests that stability in levels of social isolation over time may buffer against the negative effects of isolation (32, 33). This is because the longer adolescents remain isolated, the more they may adapt to being alone, potentially even developing positive attitudes toward this state (32, 33). The positive correlation between loneliness levels and academic performance observed in our study aligns with these findings. Nevertheless, because our study did not measure actual study time, this interpretation should be viewed with caution and verified in future research.

Concern 4: Methods sections and specifically statistics requires improvements. Considering the limited detail provided regarding normality testing, model selection for the generalized linear model, and potential confounding variables in the mediation analysis, could the authors provide a more explicit description of these methodological aspects to enhance the rigor and transparency of their findings? Additionally, given the number of statistical tests performed, did the authors consider adjustments for multiple comparisons, and if not, how might this impact the in

---

## [Decision Letter · Decision Letter 1]

19 Aug 2025

Impact of Circadian Rhythm Preference on Performance in China's Standardized Residency Training: The Mediating Role of Loneliness

PONE-D-25-16590R1

Dear Dr. Yang,

We’re pleased to inform you that your manuscript has been judged scientifically suitable for publication and will be formally accepted for publication once it meets all outstanding technical requirements.

Kind regards,

Herman Wijnen, Ph.D.

Academic Editor

PLOS ONE

Additional Editor Comments (optional):

Reviewers' comments:

Reviewer's Responses to Questions

**Comments to the Author**

1. If the authors have adequately addressed your comments raised in a previous round of review and you feel that this manuscript is now acceptable for publication, you may indicate that here to bypass the “Comments to the Author” section, enter your conflict of interest statement in the “Confidential to Editor” section, and submit your "Accept" recommendation.

Reviewer #1: All comments have been addressed

2. Is the manuscript technically sound, and do the data support the conclusions?

Reviewer #1: Yes

3. Has the statistical analysis been performed appropriately and rigorously? 

Reviewer #1: Yes

4. Have the authors made all data underlying the findings in their manuscript fully available?

Reviewer #1: No

5. Is the manuscript presented in an intelligible fashion and written in standard English?

Reviewer #1: Yes

6. Review Comments to the Author

Reviewer #1: I thank the authors for adequate answers to my questions. I have no further questions at the moment.

7. PLOS authors have the option to publish the peer review history of their article (what does this mean? ). If published, this will include your full peer review and any attached files.

**Do you want your identity to be public for this peer review?** For information about this choice, including consent withdrawal, please see our Privacy Policy .

Reviewer #1: No

---

## [Editor Report · Acceptance letter]

PONE-D-25-16590R1

PLOS ONE

Dear Dr. Yang,

I'm pleased to inform you that your manuscript has been deemed suitable for publication in PLOS ONE. Congratulations! Your manuscript is now being handed over to our production team.

Kind regards,

on behalf of

Dr. Herman Wijnen

Academic Editor

PLOS ONE